# Monovalent Salt and pH-Stimulated Gelation of Scallop (*Patinopecten yessoensis*) Male Gonad Hydrolysates/*κ*-Carrageenan

**DOI:** 10.3390/foods12193598

**Published:** 2023-09-28

**Authors:** Jia-Nan Yan, Bin Nie, Zhu-Jun Zhang, Ling-Yi Gao, Bin Lai, Ce Wang, Li-Chao Zhang, Hai-Tao Wu

**Affiliations:** 1School of Food Science and Technology, Dalian Polytechnic University, Dalian 116034, China; yjn3vv@163.com (J.-N.Y.); nbqwert2564244@163.com (B.N.); ayz02007@163.com (Z.-J.Z.); gly20020911@163.com (L.-Y.G.); bin.lai33@gmail.com (B.L.); wangceyx@163.com (C.W.); zlc@sxu.edu.cn (L.-C.Z.); 2SKL of Marine Food Processing & Safety Control, National Engineering Research Center of Seafood, Collaborative Innovation Center of Seafood Deep Processing, Dalian Polytechnic University, Dalian 116034, China; 3Institutes of Biomedical Sciences, Shanxi University, Taiyuan 030006, China

**Keywords:** scallop, *κ*-carrageenan, NaCl, KCl, responsive hydrogels

## Abstract

The gelation of scallop *Patinopecten yessoensis* male gonad hydrolysates (SMGHs) and *κ*-carrageenan (KC) subjected to pH (2–8, 3–9) and NaCl/KCl stimuli-response was investigated. SMGHs/KC gels subjected to a NaCl response exhibited an increasing storage modulus *G*′from 2028.6 to 3418.4 Pa as the pH decreased from pH 8 to 2, with corresponding *T*_23_ fluctuating from 966.40 to 365.64 *ms*. For the KCl-treated group, SMGHs/KC gels showed an even greater *G*′ from 4646.7 to 10996.5 Pa, with *T*_23_ fluctuating from 622.2 to 276.98 *ms* as the pH decreased from 9 to 3. The improved gel strength could be ascribed to the blueshift and redshift of hydroxyl groups and amide I peaks, enhanced enthalpy and peak temperature, and gathered characteristic diffraction peaks from SMGHs, KC, NaCl, and KCl. The CLSM and cryo-SEM images further reflected that SMGHs/KC gels showed more flocculation formation and denser and more homogeneous networks with smaller pore sizes in more acidic domains, especially when subjected to the KCl response. This research gives a theoretical and methodological understanding of the construction of salt- and pH-responsive SMGHs/KC hydrogels as novel functional soft biomaterials applied in food and biological fields.

## 1. Introduction

The physicochemical behavior and the structural conformation of stimuli-responsive hydrogels could significantly vary when subjected to external responses, including pH, temperature, light, ionic strength, solvent, redox, and magnetic/electric fields [1]. Stimuli-responsive hydrogels performing as “smart” materials have the advantages of great water content, 3D porous networks, and flexible shapes [2], allowing potential applications in biosensors [3], actuators [4], controlled drug delivery carriers with new hydrogel structures [5,6], tissue scaffolds [7], and regenerative medicine [8]. However, traditional stimuli-responsive hydrogels typically experience restrictions such as responding to single stimuli with resultant weak rheological strength. In this case, multi-stimuli-responsive hydrogels have been recently reported with stronger rheological properties, as represented by salep glucomannan/xanthan gum with a pH/temperature response [9], alginate/Ag/Fe_3_O_4_ with a pH/magnetic response [10], starch/alginate with a pH/amylase response [11], hyaluronan with a redox/pH response [12], and guar gum with a pH/salt response [13]. However, multi-stimuli-responsive hydrogels are still limited, and the appropriate tailoring of these hydrogels based on food materials would promise multi-stimuli-responsive, strong gel behaviors, as well as biocompatibility in polymer networks.

The scallop (*Patinopecten yessoensis*), as a type of bivalve species, has exhibited continuously increasing production since 1982, when it was first introduced in China. It contains abundant protein content with a yield of 1.79 million tons in 2022 and represents an important pillar industry associated with aquaculture. Previously, scallop *P. yessoensis* male gonad hydrolysates (SMGHs) have been demonstrated to perform as weak gels subjected to enzymolysis. It contains abundant essential amino acids, including cationic amino acids of lysine and arginine, anionic amino acids of glutamic acid and aspartic acid, and the neutral amino acid of glycine making up the major proportion. Moreover, the rheological and microstructural behaviors of SMGHs are greatly improved with the combination of *κ*-carrageenan (KC) [14], low-acyl gellan gum [15], xanthan gum, guar gum, locust bean gum, *ι*-carrageenan, sodium alginate, pectin, acacia gum, and agar [16]. Among them, SMGHs and KC reflect the most synergistic combination represented by higher gel strength and denser microstructure. In addition, KC exhibits outstanding advantages compared with other polysaccharides including gel stability, electronegativity, and thermoreversible character [17]. Furthermore, various SMGHs/polysaccharide hydrogels exhibit considerable gel performance upon diverse environmental stimuli, such as pH, mixing ratios, and different salts and strengths [18,19]. Therefore, it is essential to further construct SMGHs/KC-based multi-stimuli-responsive hydrogels with improved rheological and microstructural moieties for large-scale food and biomedical applications.

The objective of this report was to fabricate pH-/salt-stimulated SMGHs/KC hydrogels with appreciable gelation performance. The gelation characterization and mechanism were investigated by rheometry, low-field NMR (LF-NMR) relaxometry, Fourier transform infrared (FTIR), differential scanning calorimetry (DSC), X-ray diffraction (XRD), magnetic resonance imaging (MRI), and confocal laser scanning microscopy (CLSM) coupled with cryo-scanning electron microscopy (cryo-SEM). The data from this content could support the construction of SMGHs/KC-based multi-stimuli-responsive hydrogels as effective soft materials with extensively applicable pH and ionic ranges in food and biomedical fields.

## 2. Materials and Methods

### 2.1. Materials and Chemicals

Scallop *P. yessoensis* male gonads were donated by Zoneco Co., Ltd. (Dalian, China). The fresh gonads were washed, boiled to inactivate endogenous enzymes, and freeze-dried to obtain the resultant powders, which were stored at −30 °C and subjected to seal preservation before the experiment.

*κ*-Carrageenan was obtained from Aladdin Co., Ltd. (Shanghai, China). Trypsin and rhodamine B isothiocyanate (RITC) were provided by Sigma-Aldrich Co., Ltd. (St. Louis, MO, USA). NaCl and KCl were purchased from Sangon Biotech Co., Ltd. (Shanghai, China). All other chemical reagents were of analytical quality.

### 2.2. Scallop Male Gonad Hydrolysates (SMGHs) Preparation

Gonad powders were dispersed in ultrapure water with a protein content of 4% (*w*/*v*). Briefly, enzymolysis was started by trypsin addition (3000 U/g protein) at pH 8.0 and 37 °C. Then, the system was incubated at pH 8.0 with 0.5 M NaOH adjustment and 37 °C for 3 h. Finally, a 10 min boiling water bath was applied to inactivate the trypsin, and the hydrolysates were freeze-dried to obtain the corresponding powders, which were preserved at −30 °C before the experiment. SMGHs always showed negative zeta potential values of approximately around −36 mV at pH 2–9 without isoelectric points, and possessed molecular weights of 0.2–14.3 kDa, with the main fraction of 3 kDa.

### 2.3. Sample Preparation

SMGHs (50 mg/mL) and KC (13.8 mg/mL) stock solutions were separately fabricated by dissolving powders in ultrapure water, the pH of which was regulated to pH 8, 5, and 2 for NaCl-stimulated systems, and pH 9, 6, and 3 for KCl-stimulated systems. The pH was adjusted by 0.5 M NaOH and 1 M HCl. Then, SMGHs and KC were blended together with a final mass ratio of 1:1 (*w*/*w*) and a total solid content of 18 mg/mL at the same pH. After total blending, NaCl or KCl was added to the SMGHs/KC system with an eventual salt concentration of 0.2 M. The pre-SMGHs/KC gels were stored at 4 °C for 16 h to achieve sufficient hydration and gelation.

### 2.4. Rheological Measurement

The rheological properties of various pH-/salt-stimulated SMGHs/KC hydrogels were examined via a Discovery HR-1 rheometer (TA Instruments Menu Co., Ltd., New Castle, DE, USA) equipped with parallel plate geometry (d = 40 mm). In frequency sweeps, the storage modulus *G*′ and loss modulus *G*″ within 0.1–10 Hz were detected with a strain, gap value, and temperature of 0.5%, 1000 μm, and 25 °C, respectively. A 0.5% strain was obtained from the linear viscoelastic region via an oscillatory stress sweep model from 0.1 to 1000%.

### 2.5. LF-NMR Measurement

Water migration properties were detected based on an NMR analyzer (MesoMR23-060V-1, Niumag Analytical Instrument Co., Ltd., Suzhou, China) with a resonance frequency, magnetic field strength, and operating temperature of 22.4 MHz, 0.5 T, and 32 °C, respectively.

### 2.6. Magnetic Resonance Imaging (MRI) Measurement

A magnetic resonance imaging technique was employed to detect the proton density images of *T*_1_ and *T*_2_ of samples, in which the corresponding set parameters of offset slice, thickness of each layer, and slice gap were 28.3 mm, 3.0 mm, and 2.0 mm, respectively.

### 2.7. FTIR Measurement

FTIR spectra of samples were obtained using a PerkinElmer infrared spectrometer (Spectrum 100, Waltham, MA, USA). Initially, the freeze-dried sample was thoroughly blended and ground with KBr at a mass ratio of 1:100 using a mortar pestle. Then, the mixture was pressed to form a pellet using a pelleting instrument. The FTIR tests were conducted at room temperature with a wavenumber range of 4000–400 cm^−1^, spectral resolution of 4 cm^−1^, and total scans of 32.

### 2.8. DSC Measurement

The thermal properties of the samples were analyzed using DSC (DSC-60 plus, Shimadzu, Kyoto, Japan). A total of 5–10 mg of the sample was accurately weighed and placed in an aluminum crucible and sealed in aluminum pans. They were scanned between 30 and 200 °C, with a heating rate of 10 °C/min.

### 2.9. XRD Measurement

The crystal structures of the samples were measured with XRD (7000S, Shimadzu, Kyoto, Japan) with a Bragg–Brentano geometry using Cu Kα radiation. The data were collected over the 2θ range from 10° to 80° at a scanning speed of 5°/min.

### 2.10. CLSM Measurement

A Leica TCS SP8 (Leica, Wetzlar, Germany) was used to capture the CLSM images of the samples at room temperature. Initially, 1 mL of the sample (9 mg/mL) was labeled with 50 μL of RITC (1 mg/mL). Then, the stained mixture was placed on a glass bottom cell culture dish and closed with a cover slip. The incident light was excited at 561 nm and the laser beam was emitted within 550–750 nm wavelengths in helium/neon laser mode. The images were acquired with a 40× oil immersion objective.

### 2.11. Cryo-SEM Measurement

SU8010 SEM (Hitachi Co., Ltd., Tokyo, Japan) was applied to obtain cryo-SEM images. Various gels were immobilized in a copper holder and then frozen and subjected to liquid nitrogen slush, which was quickly delivered to a cryo-preparation chamber (PP3010T cryo-SEM preparation system, Quorum Technologies, Hertfordshire, UK) under vacuum. Various samples were further subjected to freeze-fracture, sublimation (−60 °C, 40 min), and Pt spraying (10 mV, 60 s). Finally, cryo-SEM images were captured at a 10.0 kV accelerating voltage.

### 2.12. Statistical Analysis

All results are presented as the mean ± standard deviation (*n* = 3). The data of LF-NMR and proton density were evaluated via Student’s *t* test, and a level of *p* < 0.05 was recognized as statistically significant.

## 3. Results and Discussion

### 3.1. Rheological Properties of pH/Salt-Responsive SMGHs/KC Hydrogels

Ion and pH are two dominant factors that induce the gelation of protein/polysaccharide composite hydrogels, mainly based on the mediation of electrostatic forces. As reported, KC is especially sensitive to monovalent salts while IC prefers divalent salts during gelation, especially in terms of viscoelastic properties of gels [20], for which reason monovalent salts were chosen for this work. In a previous study, we obtained the pH state diagram of SMGHs/KC complexes as a function of NaCl/KCl concentrations containing mixed polymers, soluble complexes, and insoluble coacervates [19]. According to the phase boundaries of pH_c_ and pH_φ1_, pH 8, 5, and 2 have been selected as representative points for hydrogels stimulated by NaCl, and pH 9, 6, and 3 have been selected as representative points for hydrogels stimulated by KCl. Different from SMGHs/KC fabricated by Yan et al. [19] only subjected to acidic pH, SMGHs/KC with 0.2 M NaCl/KCl stimulation were developed in a wide pH range from pH 9 to 2 in this work. SMGHs/KC without NaCl/KCl stimulation exhibit only relatively weak gel strength, with *G*′ ranging from 200 to 1000 Pa within pH 9–3 [18], which is significantly lower than that of SMGHs/KC gels with both salt and pH stimulation in this work.

As shown in Figure 1, SMGHs/KC hydrogels subjected to NaCl/KCl stimulation at pH 8–2/9–3 exhibited *G*′ values that always surpassed *G*′′ values and increased *G*′ as the frequency increased, representing solid-like and elastic dominant properties. In detail, SMGHs/KC hydrogels significantly increased from 2028.6 to 3418.4 Pa and from 4646.7 to 10996.5 Pa when subjected to NaCl and KCl stimulation as the pH decreased from 8 to 2 and 9 to 3, respectively. The enhanced gel strength could be ascribed to greater electrostatic attractive forces within cationic groups such as lysine and arginine in SMGHs and anionic sulfate groups (SO_4_^2−^) in KC with much more firmly crowded gel frameworks [14]. As the pH decreases, a transition from segregative to associative phase separation would occur continuously within proteins and polysaccharides. In segregative phase separation, they carry similar net charges and experience electrostatic repulsive forces, causing separation into both protein-rich and polysaccharide-rich phases. In terms of associative phase separation, the two biopolymers have opposing charges and experience electrostatic attractive forces, causing increasing protein/polysaccharide combination [21]. A similar observation has been described for Alaska pollock protein/KG [22], in which this hydrogel prefers gelation at acidic domains with higher rheological strength due to stronger electrostatic interactions between KC and proteins. As reported, a relatively alkaline environment would partially destroy hydrogen bonds in hydrogels, leading to weakened gel strength [18]. In addition, the physical combination between proteins and polysaccharides would hold soluble aggregates together and restrict the flexibility of individual biopolymer chains [23]. Additionally, increasing anionic patches on proteins at alkaline pH could also lead to stronger repulsions within biopolymers [23].

Regarding the participation of NaCl and KCl, SMGHs/KC-based hydrogels exhibited stronger gel strength with a KCl response than with a NaCl response at a similar pH, with an almost 2–3-fold increase. In general, the combination of proteins and polysaccharides involves two steps, named the Veis–Aranyi model [24]. Initially, proteins and polysaccharides possessing contrary charges could generate neutral complexes mainly governed by electrostatic forces. In addition, some nonelectrostatic forces, including hydrogen bonding, hydrophobic, and van der Waals forces, could further facilitate these complexes to rearrange and generate agglomerates, leading to increasing affinity interactions within various biopolymers [25]. Considering the dominant role of electrostatic interactions, salt ions could generate salt bridges within proteins and polysaccharides to improve the gel strength, and the salting-out effect would promote the large generation of protein particles and polysaccharide chains [19]. As reported, KC could generate a solid gel with the involvement of metal ions, especially K^+^, showing a better effect than Na^+^ [26]. A similar phenomenon has been reported by Haug et al. [27] where fish gelatin/KC mixtures could exhibit a maximum gel strength of approximately 25 kPa with 20 mM KCl addition but a lower gel strength of less than 5 kPa with 20 mM NaCl addition. Indeed, K^+^ is specifically bound to KC, while Na^+^ only affects the carrageenan network via common ionic effects, as the salt-enhanced effect is much more noticeable with K^+^ than with Na^+^ for sulfated polysaccharides [27]. Accordingly, rheological properties have a good correlation with mechanical properties, in which good rheological strength reflects excellent mechanical strength. This correlation has been reported in soy protein/chitin [28], oat protein/polysaccharides (carrageenan, dextrin, and chitosan) [29], and chitosan/alginate/fucoidan complex gels [30]. Therefore, it is suggested that an acidic medium combined with KCl stimulation is highly favorable for SMGHs/KC hydrogel construction due to stronger electrostatic interactions within SMGHs and KC, and more K^+^ in KC junction zones for specific binding.

### 3.2. Water Migration and Distribution of pH/Salt-Responsive SMGHs/KC Hydrogels

The *T*_2_ relaxation time of pH/salt-responsive SMGHs/KC hydrogels is presented in Figure 2A,B, and Appendix A. Overall, two obvious peaks appeared at 0.53–0.70 *ms* (*T*_21_) and 276.98–1086.11 *ms* (*T*_23_) in the SMGHs/KC gels subjected to pH/salt-responsive stimuli (Figure 2A,B). Typically, *T*_21_ represents bound water closely combined with macromolecules, and free water is located in the exterior of the gel framework with more flexibility [14]. As presented, the fraction of *T*_23_ was detected to be more than 98%, indicating the predominant part of free water in SMGHs/KC gels subjected to pH/salt-stimuli-response. Moreover, as pH decreased, *T*_21_ was almost unaffected by pH, while *T*_23_ initially increased from 966.40 to 1035.88 *ms* for the NaCl group and 622.26 to 1086.11 *ms* for the KCl group, and then dramatically decreased to 365.64 *ms* and 276.98 *ms*, respectively (*p* < 0.05) (Figure 2A,B). Accordingly, the *T*_2_ relaxation time could indicate the water dynamic state in protein-derived gels in situ with no external solution removal [14], in which shorter relaxation times indicate tighter water–biopolymer interactions with higher gel strength. In this work, SMGHs/KC gels reflected a shorter *T*_23_ at a more acidic domain and with a KCl response, corresponding well to the rheological data that SMGHs/KC gels exhibited the highest *G*′ values of even 10,996.5 Pa when subjected to pH 3 and 0.2 M KCl (Figure 1 and Figure 2B,C). To some extent, with the response of acidic pH and KCl, SMGHs/KC gels tended to limit the flexibility of water molecules and enhance the binding strength of water in the gel [14]. Therefore, it is suggested that *T*_23_ undergoes a fluctuation as the pH drops from 8 to 2 and 9 to 3 in pH/salt-responsive SMGHs/KC hydrogels, in which SMGHs/KC with KCl and more acidic pH responses could restrict water mobility and therefore support stronger rheological strength.

*T*_1_ and *T*_2_ proton density images relating to the bound and free water distribution of pH/salt-stimuli-responsive SMGHs/KC hydrogels are presented in Figure 2C,D. Accordingly, the red color represents a high proton signal density with more hydrogen protons and the blue color represents a low proton signal density [31]. Obviously, the water was unevenly spatially distributed within the pH/salt-responsive SMGHs/KC hydrogels, and the *T*_2_ group reflected a brighter and redder signal density than the *T*_1_ group (Figure 2A,B), indicating the dominant part of free water in the composite gels. However, when comparing *T*_1_ and *T*_2_ weighted images separately, they showed comparable quantitative signal intensity with no significant difference when subjected to various pH and salt responses. Thus, it could be assumed that the water distribution in SMGHs/KC gels is steadily subjected to pH and NaCl/KCl responses, and free water occupies the dominant part in the composite gels with denser hydrogen protons.

### 3.3. FTIR Spectra of pH/Salt-Responsive SMGHs/KC Hydrogels

The intermolecular interactions within SMGHs and KC subjected to various pH values and salts were characterized by FTIR (Figure 3A). Obviously, pH and salts could mainly change the O–H stretching vibration of hydroxyl groups and the absorption peaks of amide I bands. The original spectra of KC and SMGHs showed O–H and amide I peaks at 3444 and 1642 cm^−1^ and 3300 and 1647 cm^−1^, respectively (Figure 3A). With the stimulation of pH and salt, SMGHs/KC exhibited corresponding peaks within 3300 and 3444 cm^−1^, and 1642 and 1647 cm^−1^, respectively (Figure 3A), indicating the good compatibility and interaction between SMGHs and KC. A similar phenomenon has also been found by Guo [32] where FTIR peaks of pea protein/high methoxyl pectin complexes are composed of representative peaks in single protein and polysaccharide, speculating the effective interactions existing in the complexes. Moreover, as the pH decreased from 8 to 2, the O–H stretching bands in NaCl-responsive SMGHs/KC showed a decreasing trend (3435–3428 cm^−1^), while the amide I peaks showed an increasing trend (1637–1647 cm^−1^) (Figure 3A). However, O–H stretching bands and amide I peaks in NaCl-responsive SMGHs/KC gels showed a contrary tendency (Figure 3A). As reported, the blueshifts of O–H stretching bands and amide I peaks represent stronger hydrogen bonds and electrostatic interactions, respectively [33]. Thus, it is assumed that hydrogen bonds are more dominant in NaCl-stimulated SMGHs/KC hydrogels while electrostatic interactions are more favorable for KCl-stimulated SMGHs/KC hydrogels as the pH decreases.

### 3.4. DSC Curves of pH/Salt-Responsive SMGHs/KC Hydrogels

The DSC thermograms of various pH/salt-responsive SMGHs/KC hydrogels are presented in Figure 3B. All samples showed a broad endothermic peak characteristic during heating. Single SMGHs and KC exhibited an endothermal peak at approximately 84.3 °C and 104.9 °C, respectively. With the stimulation of pH/salt, SMGHs/KC hydrogels exhibited a relatively lower endothermal peak of approximately 63 °C and enthalpy of approximately 152.7 J/g upon NaCl treatment and 76.2 °C combined with 213.8 J/g upon KCl treatment. Obviously, in comparison to the NaCl-treated group, KCl significantly increased the denaturation temperature and enthalpy of SMGHs/KC hydrogels. This result indicated that stimulation with KCl greatly increased the thermal stability of SMGHs/KC hydrogels compared with NaCl stimulation. These results were consistent with rheological data showing that KCl was more favorable for SMGHs/KC hydrogel construction. Therefore, it is suggested from a thermodynamic point of view that the beneficial effect of KCl could increase the difficulty of destroying the internal structure of SMGHs/KC at high temperatures.

### 3.5. XRD Spectra of pH/Salt-Responsive SMGHs/KC Hydrogels

XRD measurements were applied to investigate the changes in the crystallinity of the pH/salt-responsive SMGHs/KC mixtures. SMGHs and KC exhibited one wider hump at 2θ values of approximately 20° and 21° (Figure 3C), indicating the amorphous nature of the SMGHs and KC. NaCl and KCl presented characteristic diffraction peaks at 31.7 and 66.2°, 28.3°, and 58.6° (Figure 3C), respectively, indicating the crystalline structure of salts. Collectively, various pH/salt-responsive SMGHs/KC mixtures exhibited amorphous structures with broad peaks in the vicinity of 28.3° and 50.2°, as well as some narrow peaks, consistent with those of SMGHs, KC, and salts. These results indicated that SMGHs and KC might have good compatibility and interaction, thus modifying the original crystal structure of SMGHs/KC. A similar phenomenon has also been found by Guo et al. [32] where crystallization peaks of CaCl_2_-induced pea protein/high methoxyl pectin complexes are consistent with those of single pea protein, high methoxyl pectin, and CaCl_2_. Thus, it is assumed that SMGHs and KC, upon pH/salt-responsiveness, have good compatibility in complex gel systems, maintained by intermolecular interactions.

### 3.6. Microstructural Properties of pH/Salt-Responsive SMGHs/KC Hydrogels

Confocal images of pH/salt-responsive SMGHs/KC hydrogels are shown in Figure 4A. As the pH decreased, SMGHs were gradually protonated, leading to more SMGH aggregation. Moreover, increasing cationic SMGH patches containing lysine and arginine could electrostatically interact with anionic sulfate groups in KC, leading to more SMGHs/KC coacervates. Collectively, the denser and larger SMGH aggregates and SMGHs/KC coacervates led to continuous, homogeneous, and dispersed flocculation, coupled with the formation of salt bridges and phase separation. These phenomena could be ascribed to greater binding between SMGHs and KC with the stimulation of pH and salt, especially KCl, as reflected by rheological, water migration, FTIR, and DSC data. Therefore, it is suggested that a more acidic pH combined with NaCl/KCl stimulation would be beneficial to the flocculation of SMGHs/KC, as revealed by more RITC staining corresponding to diameter and density, which are favorable for dramatically enhanced rheological properties.

The cryo-SEM images of pH/salt-stimuli-responsive SMGHs/KC hydrogels are shown in Figure 4B. Overall, as the pH decreased, the network of SMGHs/KC gels became denser with decreased pore sizes and thicker network walls, exhibiting a honeycomb-like well-distributed network. Moreover, in comparison to NaCl stimulation, KCl stimulation could induce SMGHs/KC gels with even smaller pore sizes and homogeneous networks. In general, the pore size of the gel network is well associated with the interaction between biopolymers and water, in which smaller pore sizes could contribute to stronger water–biopolymer interactions, with less water leaving [34]. Additionally, the smaller pore size would cause a more specific surface of the gel skeletal architecture, allowing a smaller hydration space in which SMGHs/KC and water molecules could closely combine. Thus, water in SMGHs/KC gels would hardly migrate at lower pH and KCl stimulation due to smaller pores, leading to stronger cross-linking within SMGHs and KC with enhanced gel toughness and less *T*_2_ relaxation time as shown in Figure 1 and Figure 2A,B. In contrast to our observation, Yan et al. [35] have observed that the microstructures of corn fiber gum/soy protein hydrogels convert from coarse and irregular to smooth and ordered as the pH increases from 5.0 to 7.5 with excellent textural properties. Indeed, as the pH is distant from the pI (4.5–5.2) of soy protein, the increase in the net charges on the surface of the two polymers could enhance interactions between corn fiber gum and soy protein, contributing to the formation of more stable and regular networks. Collectively, SMGHs/KC gels could generate denser and more well-distributed networks subjected to more acidic pH and KCl responses, and thereby energetically lock more water to construct gel networks with greater rheological strength.

## 4. Conclusions

SMGHs/KC gels were constructed and subjected to pH (2–8, 3–9) and NaCl/KCl stimuli-response. Collectively, SMGHs/KC gels exhibited stronger rheological strength and shortened water migration in an acidic medium due to stronger electrostatic interactions between SMGHs and KC. In addition, the KCl stimulus response contributed to higher rheological strength and thermal stability because of more K^+^ in KC junction zones for specific binding. Moreover, water distribution in SMGHs/KC gels was steadily subjected to pH and NaCl/KCl response, and free water occupied the dominant part with denser hydrogen protons. Hydrogen bonds were more dominant in NaCl-stimulated SMGHs/KC hydrogels while electrostatic interactions were more favorable for KCl-stimulated SMGHs/KC hydrogels as the pH decreased, leading to good compatibility in gelation between SMGHs and KC. Furthermore, SMGHs/KC gels exhibited denser and more homogeneous networks with more flocculation formation when subjected to more acidic pH and KCl responses to support gel frameworks with stronger gel strength. The current work proposed that SMGHs/KC gels could be developed as potential materials with appreciable rheological strength and microstructural networks. Therefore, they contribute to the development of encapsulation of bioactive components, modification of textural and sensory properties, fabrication of edible and antibacterial coatings, and construction of fat replacers in food and biological fields.

## Figures and Tables

**Figure 1 foods-12-03598-f001:**
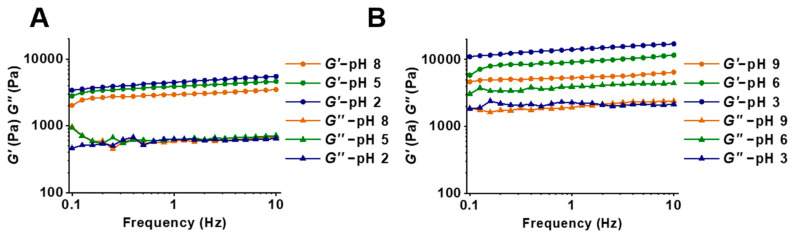
Rheological profiles of SMGHs/KC hydrogels in response to pH/salt stimuli. (**A**) Storage modulus *G*′ and loss modulus *G*″ curves of SMGHs/KC gels subjected to pH 2–8 and 0.2 M NaCl stimuli. (**B**) Storage modulus *G*′ and loss modulus *G*″ curves of SMGHs/KC gels subjected to pH 3–9 and 0.2 M KCl stimuli.

**Figure 2 foods-12-03598-f002:**
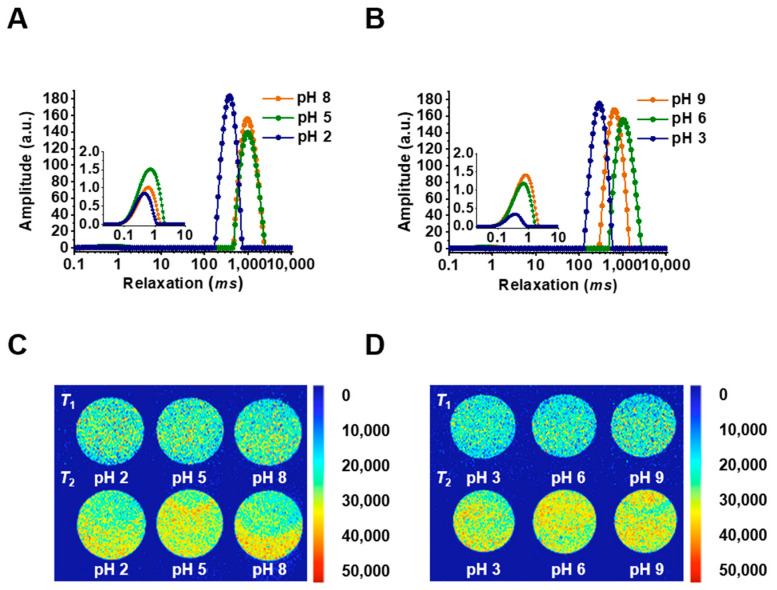
Water migration and water distribution profiles of SMGHs/KC hydrogels in response to pH/salt stimuli. (**A**) *T*_2_ relaxation time curves of SMGHs/KC gels at pH 2–8 and 0.2 M NaCl. (**B**) *T*_2_ relaxation time curves of SMGHs/KC gels at pH 3–9 and 0.2 M KCl. (**C**) *T*_1_ and *T*_2_ weighted images of SMGHs/KC gels upon pH 2–8 and 0.2 M NaCl stimulus response. (**D**) *T*_1_ and *T*_2_ weighted images of SMGHs/KC gels upon pH 3–9 and 0.2 M KCl stimulus response.

**Figure 3 foods-12-03598-f003:**
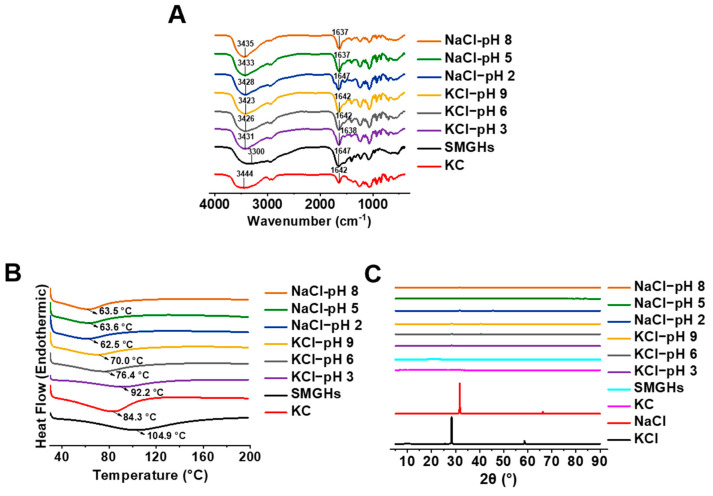
Conformation characterization of SMGHs/KC mixtures in response to pH/salt stimuli. (**A**) FTIR curves, with SMGHs and KC as controls. (**B**) DSC curves, with SMGHs and KC as controls. (**C**) XRD curves, with SMGHs, KC, NaCl, and KCl as controls.

**Figure 4 foods-12-03598-f004:**
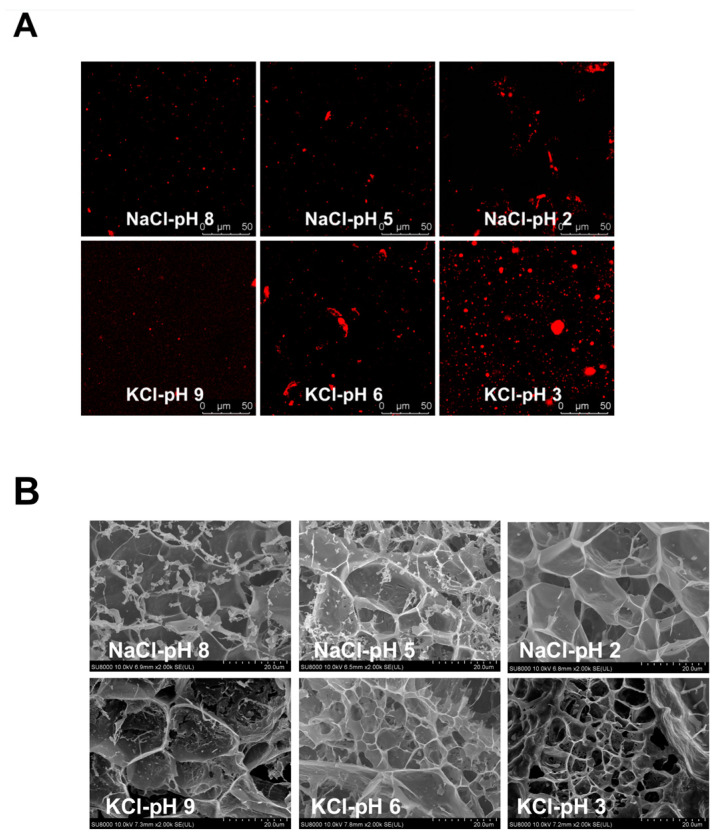
Microstructural images of SMGHs/KC hydrogels in response to pH/salt stimuli. (**A**) CLSM micrographs of SMGHs/KC gels upon pH 2–8 and 0.2 M NaCl stimulus response. (**B**) Cryo-SEM micrographs of SMGHs/KC gels upon pH 3–9 and 0.2 M KCl stimulus response.

## Data Availability

The data presented in this study are available on request from the corresponding author.

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
