# Peer review of "Monovalent Salt and pH-Stimulated Gelation of Scallop (Patinopecten yessoensis) Male Gonad Hydrolysates/κ-Carrageenan"

_foods, 2023, doi:10.3390/foods12193598_

Round 1

Reviewer 1 Report

The paper contains reports in production of pH-/salt-stimulated scallop (Pati-2 nopecten yessoensis) male gonad hydrolysates/κ-carrageenan gel. The structure of the manuscript needs revision, and some points should be taken into account and corrected:

1- Title: It is good.

 2- Abstract It can be improved.

Line 16- 18: it is too long and confusing.

3- Keywords: They were well chosen.

4- Introduction relatively well written, you can add recent investigation about developing new hydrogel structure, such as : Development of soy protein/sodium alginate nanogel-based cress seed gum hydrogel for oral delivery of curcumin

Line 55- Why was Carrageenan selected for creating the gel, please add som explantion about this polysaccharide.

5- Material and methods:

Sample characterization needs revision, each method should have subtitle and explain with more details.

Rheological test:

Mention to the name of test: frequency test, the type of spindle, why 0.5 % strain was chosen, the LVE test should be mentioned.

CLSM test needs more explanation. 

Line 114. Which data was analyzed based on t-test, no comparison was observed.

Results:

English usage in this manuscript must be substantially improved. There are many grammatical errors and vague descriptions, especially in results and discussion. Th sentences are too long.

Line 124-136: the change in protein surface charge with pH change can influence on electrostatic interaction and change soluble interaction to insoluble interaction. This part needs more explanation.

Why monovalent salt was used, divalent salt have more effect, especially on viscoelastic properties of gel.

Line 212- what do you mean: flocculation? Do you mean aggregation?

-

Author Response

Thank you for your e-mail of August 23, 2023, from which we are pleased to learn that Foods is interested, in principle, in publishing our manuscript entitled ‘Monovalent salt and pH-stimulated gelation of scallop (Patinopecten yessoensis) male gonad hydrolysates/κ-carrageenan’. We have studied the reviewer’s comments very carefully and have made necessary corrections according to the suggestions. The comments offered by the reviewers have been helpful in formulating our manuscript. We deeply appreciate these thoughtful comments, and hope that our responses and in particular our revisions have allowed this paper to achieve a priority sufficient for publication in Foods. The revisions are highlighted with red ink and underline in the revised manuscript.

Response to Reviewer #1:

Thank you for the reviewer for his/her careful reading of the manuscript and constructive suggestions. We have revised the manuscript according to the suggestions of the reviewer. We believe that the paper is improved and hope that the corrections made together with the attached reply satisfy the concerns raised.

1) Abstract: It can be improved. Line 16- 18: it is too long and confusing.

As suggested, we have improved the Abstract by describing NaCl-treated and KCl-treated SMGHs/KC hydrogels separately and make the sentence short and clear (Lines15-19).

2) Introduction relatively well written, you can add recent investigation about developing new hydrogel structure, such as: Development of soy protein/sodium alginate nanogel-based cress seed gum hydrogel for oral delivery of curcumin.

As suggested, we have added recent investigation about  soy protein/sodium alginate nanogel-based cress seed gum hydrogel (Line 38), and cited one new reference in the References Section (Lines 449-451).

3) Line 55- Why was Carrageenan selected for creating the gel, please add some explantion about this polysaccharide.

As suggested, we have added some explanation about carrageenan for creating the gel (Lines 61-65), and cited one new reference in the References Section (Lines 481-482).

4) Rheological test:

Mention to the name of test: frequency test, the type of spindle, why 0.5 % strain was chosen, the LVE test should be mentioned.

As suggested, we have added more details on rheological test (Lines108, 111, 113-115).

5) CLSM test needs more explanation.

As suggested, we have added more details on CLSM test (Lines 142, 145-146).

6) Line 114. Which data was analyzed based on t-test, no comparison was observed.

As suggested, the data of LF-NMR and proton density data was analyzed based on t-test and we have provided these data in Tables S1 and S2 (Lines 154-156).

7) English usage in this manuscript must be substantially improved. There are many grammatical errors and vague descriptions, especially in results and discussion. The sentences are too long.

As suggested, we have checked throughout the manuscript and improved English usage in this manuscript including grammatical errors and vague descriptions (Lines 15-18, 36-37, 41, 43-45, 95, 111-112, 154, 180, 183, 210-211, 221-223, 226, 240, 245, 255, 270-271, 276, 366, 373, 378, and 391), and shortened some sentences (Lines 15-19, 32, 35-38, 55-58, 61-63, 72-75, 179-180, 182-186, 239, 253, 255-258, 266-268, 342-344, and 389-391).

8) Line 124-136: the change in protein surface charge with pH change can influence on electrostatic interaction and change soluble interaction to insoluble interaction. This part needs more explanation.

As suggested, we have provided more explanation of the transition from segregative to associative phase separation (Lines 187-193), and cited one new reference in the References Section (Lines 495-495).

9) Why monovalent salt was used, divalent salt have more effect, especially on viscoelastic properties of gel.

We agree to the reviewer’s idea that we should explain why monovalent salt was used clearly. To this point, we have added the details in the Results and discussion section (Lines 165-167), and cited one new reference in the References Section (Lines 489-491).

10) Line 212- what do you mean: flocculation? Do you mean aggregation?

As suggested, we have given the meaning of flocculation, which was comprised of SMGHs aggregates and SMGHs/KC coacervates (Lines 336-340).

Reviewer 2 Report

Dear Authors,

Please consider the changing some sentences - there are too long and it is difficult to understand what is the main idea.

Why did you choose different pH in case of hydrogels stimulated by KCl and NaCl? It is difficult to compare them.

There is lack of contol hydrogel. Please add the controls. Even new hydrogels are worth comparing them with control hydrogels.

Dear Authors,

English language correction is needed (some sentenses are too long).

Author Response

Thank you for your e-mail of August 23, 2023, from which we are pleased to learn that Foods is interested, in principle, in publishing our manuscript entitled ‘Monovalent salt and pH-stimulated gelation of scallop (Patinopecten yessoensis) male gonad hydrolysates/κ-carrageenan’. We have studied the reviewer’s comments very carefully and have made necessary corrections according to the suggestions. The comments offered by the reviewers have been helpful in formulating our manuscript. We deeply appreciate these thoughtful comments, and hope that our responses and in particular our revisions have allowed this paper to achieve a priority sufficient for publication in Foods. The revisions are highlighted with red ink and underline in the revised manuscript.

Response to Reviewer #2:

Thank you for the reviewer for his/her careful reading of the manuscript and constructive suggestions. We have revised the manuscript according to the suggestions of the reviewer. We believe that the paper is improved and hope that the corrections made together with the attached reply satisfy the concerns raised.

1) Please consider the changing some sentences - there are too long and it is difficult to understand what is the main idea.

As suggested, we have changed the long sentences to shorter one (Lines 15-19, 32, 35-38, 55-58, 61-63, 72-75, 179-180, 182-186, 239, 253, 255-258, 266-268, 342-344, and 389-391).

2) Why did you choose different pH in case of hydrogels stimulated by KCl and NaCl? It is difficult to compare them.

we agree to the reviewer’s idea that we should explain why we chosen different pH in case of hydrogels stimulated by KCl and NaCl. To this point, we have revised the statement and added more detail in the Results and Discussion Section (Lines 167-172).

3) There is lack of contol hydrogel. Please add the controls. Even new hydrogels are worth comparing them with control hydrogels.

As suggested, we have provided the rheological data of SMGHs/KC control hydrogels without KCl/NaCl stimulation within a wide pH range of 9-3. Furthermore, we have revised the statement and added more detail in the Results and Discussion Section (Lines 175-178).

4) English language correction is needed (some sentenses are too long).

As suggested, we have corrected English language (Lines 15-18, 36-37, 41, 43-45, 95, 111-112, 154, 180, 183, 210-211, 221-223, 226, 240, 245, 255, 270-271, 276, 366, 373, 378, and 391) and make some sentences shorter (Lines 15-19, 32, 35-38, 55-58, 61-63, 72-75, 179-180, 182-186, 239, 253, 255-258, 266-268, 342-344, and 389-391).

Reviewer 3 Report

The article submitted for review is devoted to the study of gelling and properties of hydrogels formed by hydrolysates of scallop male gonad hydrolysates and κ-carrageenan at various pH in the presence of NaCl or KCl. The rheological properties and microstructure of the hydrogels have been studied. Such gels have significant potential for use in food technology and biomedicine.

Reading the manuscript raises several questions.

1. It would be good to give the amino acid composition of the studied hydrolysates, which would show the presence of cationic groups.

2. How did you set the pH values in the system when preparing the samples?

3. Based on the increase in storage modulus, the authors infer “greater electrostatic attractive forces within cationic groups in SMGHs and anionic sulfate groups in KC with much more firmly crowded gel frameworks” (lines 126-128). But this is not an obvious conclusion.

4. To explain the effect of calcium and sodium cations on the elastic-viscous properties of hydrogels, it would be nice to have data for hydrogels formed without the addition of salts.

5. Line 206. The authors write: “gels gradually began to generate denser and larger aggregates, leading to continuous, homogeneous and dispersed flocculation” – it is necessary to clarify what the authors mean by aggregates and homogeneous flocculation.

6. Further, the authors write (line 210), “with the formation of salt bridges and phase separation, which was ascribed to the greater binding between SMGHs and KC”. How do the authors measure  "stronger binding"?

7. What do the authors mean by "improved gelling properties" (line 213-214)?

8. In conclusion, the authors write about “stronger mechanical strength” (line 243). But the mechanical strength of the hydrogels was not determined in the work.

9. It is not obvious that “SMGHs/KC gels” are “novel materials” (line 251).

10. The list of references should be expanded. It should not be limited mainly to the work done by the compatriots of the authors of the article under review.

In conclusion, it should be noted that the article presents experimental data that are consistent with each other, but their interpretation requires significant adjustment.

Author Response

Thank you for your e-mail of August 23, 2023, from which we are pleased to learn that Foods is interested, in principle, in publishing our manuscript entitled ‘Monovalent salt and pH-stimulated gelation of scallop (Patinopecten yessoensis) male gonad hydrolysates/κ-carrageenan’. We have studied the reviewer’s comments very carefully and have made necessary corrections according to the suggestions. The comments offered by the reviewers have been helpful in formulating our manuscript. We deeply appreciate these thoughtful comments, and hope that our responses and in particular our revisions have allowed this paper to achieve a priority sufficient for publication in Foods. The revisions are highlighted with red ink and underline in the revised manuscript.

Response to Reviewer #3:

Thank you for the reviewer for his/her careful reading of the manuscript and constructive suggestions. We have revised the manuscript according to the suggestions of the reviewer. We believe that the paper is improved and hope that the corrections made together with the attached reply satisfy the concerns raised.

1) It would be good to give the amino acid composition of the studied hydrolysates, which would show the presence of cationic groups.

As suggested, we have given the amino acid composition of the studied hydrolysates in the Introduction Section (Lines 55-58).

2) How did you set the pH values in the system when preparing the samples?

As suggested, we have explained how set the pH values in the system when preparing the samples (Lines 103, 167-172).

3) Based on the increase in storage modulus, the authors infer “greater electrostatic attractive forces within cationic groups in SMGHs and anionic sulfate groups in KC with much more firmly crowded gel frameworks” (lines 126-128). But this is not an obvious conclusion.

We agreed to reviewers idea that “greater electrostatic attractive forces within cationic groups in SMGHs and anionic sulfate groups in KC with much more firmly crowded gel frameworks” is not an obvious conclusion.

As suggested, we have given a more obvious conclusion (Lines 182-186).

4) To explain the effect of calcium and sodium cations on the elastic-viscous properties of hydrogels, it would be nice to have data for hydrogels formed without the addition of salts.

As suggested, we have provided data for hydrogels formed without the addition of salts (Lines 175-178).

5) Line 206. The authors write: “gels gradually began to generate denser and larger aggregates, leading to continuous, homogeneous and dispersed flocculation” – it is necessary to clarify what the authors mean by aggregates and homogeneous flocculation.

As suggested, we have clarified aggregates and homogeneous flocculation in the Results and Discussion Section (Lines 336-340).

6) Further, the authors write (line 210), “with the formation of salt bridges and phase separation, which was ascribed to the greater binding between SMGHs and KC”. How do the authors measure "?

We agreed to reviewers idea that “stronger binding” between SMGHs and KC should be measured. we have measured stronger binding based on rheological, water migration, FTIR and DSC data.

To this point, we have revised the statement and added more detail in the Abstract Section (Lines 19-20), Introduction Section (Lines 74-75), Materials and Methods Section (Line 126-137), Results and Discussion Section (Lines 278-317), and Conclusion Section (Lines 383-386), and cited one new reference in the References Section (Lines 520-522).

7) What do the authors mean by "improved gelling properties" (line 213-214)?

We agreed to reviewers idea that we should give the mean of “improved gelling properties". We have given details on “improved gelling properties" (Line 347).

8) In conclusion, the authors write about “stronger mechanical strength” (line 243). But the mechanical strength of the hydrogels was not determined in the work.

We agreed to reviewers idea that the mechanical strength of the hydrogels was not determined in the work. In this case, we have provide the correlation between rheological strength and mechanical strength, and replaced the “mechanical strength” as “rheological strength” throughout the manuscript.

To this point, we have revised the statement and added more detail in the Introduction Section (41-42, 69), Results and Discussion Section (Lines 196, 258, 374), and Conclusion Section (Lines 377,391), and cited three new reference in the References Section (Lines 511-519).

9) It is not obvious that “SMGHs/KC gels” are “novel materials” (line 251).

As suggested, we have changed “novel materials” to “potential materials” (Line 390).

10) The list of references should be expanded. It should not be limited mainly to the work done by the compatriots of the authors of the article under review.

As suggested, we have expanded the list of references (Lines 434-436, 481-482, 489-491, 495-496, and 511-522).

Reviewer 4 Report

The authors reported the preparation of hydrogels based on scallop male gonad hydrolysates to study the gelation stimulated under two different salts (NaCl and KCl) and different pH values (8 to 2 and 9 to 3). The gelation performance was evaluated by means of rheometry, low-field NMR, CLMS and cryo-SEM.

It was difficult to appreciate the contribution of this manuscript when compared with those works described in bibliography. As an example we can mention: Yan, J.N.; Nie, B, Jiang, X.Y.; Han, J.R.; Du, Y.N.; Wu, H.T. Complex coacervation of scallop (Patinopecten yessoensis) male gonad hydrolysates and κ-carrageenan: Effect of NaCl and KCl. Food Res. Int. 2020, 137, 109659. https://doi.org/10.1016/j.food-res.2020.109659.

Some issues could be mentioned once both papers were analysed:

-        The gelation behaviour of the hydrogels was studied with the same raw materials (SMGH) and biopolymer such as carrageenan with the same salts (NaCl/KCl).

-        The same techniques were applied to experimental work. It included: Rheology, relaxation times, microscopy images.

-        Therefore, the conclusion section resulted quite similar to previous paper

-        I strongly recommend to re-write the manuscript with new experimental data obtained with different salts, different techniques that could elucidate the gelation mechanism under different perspectives. You can include several methods such as Zeta potential, isoelectric point, mechanical and thermal properties, X ray, FTIR, Mw of hydrolysates, etc.

Minor editing of English language required

Author Response

Thank you for your e-mail of August 23, 2023, from which we are pleased to learn that Foods is interested, in principle, in publishing our manuscript entitled ‘Monovalent salt and pH-stimulated gelation of scallop (Patinopecten yessoensis) male gonad hydrolysates/κ-carrageenan’. We have studied the reviewer’s comments very carefully and have made necessary corrections according to the suggestions. The comments offered by the reviewers have been helpful in formulating our manuscript. We deeply appreciate these thoughtful comments, and hope that our responses and in particular our revisions have allowed this paper to achieve a priority sufficient for publication in Foods. The revisions are highlighted with red ink and underline in the revised manuscript.

Response to Reviewer #4:

Thank you for the reviewer for his/her careful reading of the manuscript and constructive suggestions. We have revised the manuscript according to the suggestions of the reviewer. We believe that the paper is improved and hope that the corrections made together with the attached reply satisfy the concerns raised.

1) The gelation behaviour of the hydrogels was studied with the same raw materials (SMGH) and biopolymer such as carrageenan with the same salts (NaCl/KCl).

As suggested, we have clarified the differences between materials used in previous and current study (Lines 173-175).

2) The same techniques were applied to experimental work. It included: Rheology, relaxation times, microscopy images.

As suggested, we have provided several new methods including DSC, XRD and FTIR (Lines 126-141).

3) Therefore, the conclusion section resulted quite similar to previous paper.

As suggested, we have modified the conclusion Section being different from previous paper (Lines 377, 379-380, 383-387, and 389-391).

4) I strongly recommend to re-write the manuscript with new experimental data obtained with different salts, different techniques that could elucidate the gelation mechanism under different perspectives. You can include several methods such as Zeta potential, isoelectric point, mechanical and thermal properties, X ray, FTIR, Mw of hydrolysates, etc.

As suggested, we have included several methods such as mechanical and thermal properties, X ray, and FTIR for pH/salt-responsive SMGHs/KC hydrogels. Moreover, we have provided some information relating to Zeta potential, isoelectric point, and Mw of SMGHs hydrolysates, etc to elucidate the gelation mechanism under different perspectives.

To this point, we have modified the title, revised the statement and added more detail in the Abstract Section (Lines 19-21), Introduction Section (Lines 74-75), Materials and Methods Section (Lines 96-98, 126-141, and 158-159), Results and Discussion Section (Lines 227-231, 279-333, and 342-344), and Conclusion Section (Lines 383-386), and cited two new reference in the References Section (Lines 511-522).

Round 2

Reviewer 1 Report

The manuscript has been improved, especially in the results and discussion section, and it is acceptable in present form.

-

Reviewer 2 Report

Dear Authors,

I accept your explenations and corrections that have been made.

Reviewer 4 Report

The recommendations were attended satisfactorily. There were included several techniques for the characterization of the samples and the elucidation of the gelation mechanism.